# Assessing the Prevalence and Dynamics of Emerging Campylobacterales in Human Stool Samples in Brussels by Filtration Culture

**DOI:** 10.3390/pathogens13060475

**Published:** 2024-06-04

**Authors:** Emmanuelle Giraudon, V. Y. Miendje Deyi, Delphine Martiny

**Affiliations:** 1Department of Microbiology, Laboratoire Hospitalier Universitaire de Bruxelles-Brussel Universitair Laboratorium (LHUB-ULB), Université Libre de Bruxelles (ULB), 1000 Brussels, Belgiumdelphine.martiny@lhub-ulb.be (D.M.); 2Belgium National Reference Center for Campylobacter (LHUB-ULB), 1000 Brussels, Belgium; 3Faculty of Medicine and Pharmacy, University of Mons (UMONS), 7000 Mons, Belgium

**Keywords:** emerging Campylobacter, fastidious Campylobacter, *C. ureolyticus*, gastroenteritis, filtration culture, stool samples, epidemiology, seasonal prevalence, age distribution, Belgium

## Abstract

Thermophilic *C. jejuni*/*coli* is reported to be the first bacterial cause of gastroenteritis worldwide and the most common zoonosis in Europe. Although non-*jejuni*/*coli Campylobacter* sp. are increasingly suspected to be responsible for diarrhoea or to be involved in inflammatory bowel disease, they remain poorly isolated due to their fastidious and non-thermophilic nature. Additionally, they are not targeted by commercial syndromic PCR assays. In this study, we present routine diagnostic results over 6 years (2017–2019 and 2021–2023) of *Campylobacter* sp. and related species, obtained by optimised culture from 51,065 stools by both 0.65 µm pore filtration on antibiotic-free agar, incubated in an H_2_-enriched atmosphere at 37 °C (also known as the Cape Town protocol), and the use of selective inhibitory Butzler medium incubated at 42 °C. This allowed the isolation of 16 *Campylobacter* species, 2 *Aliarcobacter* species, and 2 *Helicobacter* species, providing a completely different view of the epidemiology of Campylobacterales, in which *C. jejuni*/*coli* represents only 30.0% of all isolates, while *C. concisus* represents 44.4%. *C. ureolyticus*, representing only 5.5% of all Campylobacterales pre-COVID-19, represented 20.6% of all strains post-COVID-19 (218% increase; *p* < 0.05). At the same time, the proportions of *C. jejuni*, *C. coli*, and *C. concisus* decreased by 37, 53, and 28%, respectively (*p* < 0.05).

## 1. Introduction

*Campylobacter* sp. diarrhoea was first described by T. Escherich in 1886 [1] as “cholera infantum” in newborns. Then, in the first half of the 20th century, “vibrio-like” organisms were isolated in miscarrying sheep, cattle, and ewes [2,3,4,5,6,7,8,9,10,11,12], or calves and swine with diarrhoea [13,14,15,16,17]. It was not until 1947 that Vinzent et al. [18] reported a case of fatal “Vibrion” septicaemia in a pregnant woman, followed by case reports of “related Vibrio” diarrhoea, particularly in children [19,20,21,22,23]. This led to the creation of the genus *Campylobacter* by Sebald and Veron in 1963 [24]. *Vibrio jejuni* and *V. coli* were eventually renamed *C. jejuni* and *C. coli*, and in 1972 Dekeyser and Butzler [25] first isolated *Campylobacter* sp. from stool samples of diarrhoeal patients.

Subsequently, in the 1970s, two methods were proposed for the selective culture of *Campylobacter* sp. from stools: culture on inhibitory medium containing antibiotics [26,27,28,29,30,31,32,33,34,35,36,37,38,39,40,41,42,43] or culture on non-selective enriched medium after selective filtration of stools [32,38,44,45,46,47,48,49,50,51,52,53,54,55,56,57,58,59,60,61,62,63,64], also called “The Cape Town protocol” [50]. Filtration techniques allowed to isolate non-*jejuni/coli Campylobacter* sp., especially *C. concisus* [40,51,52,55,56,57,59,60,61,63,64], *C. curvus* [40,51,55,56,57,59,61,64,65], *C. ureolyticus* [59,61,63,64], and *C. upsaliensis* [45,51,52,54,55,56,57,59,60,61,64] (Appendix A). The best results were obtained using polycarbonate instead of cellulose filters [60,63], with pore sizes of 0.6 µm instead of 0.45 µm [58,63,64], and incubating plates in H_2_-enriched microaerobic atmosphere [53,54,64] at 37 °C instead of 42 °C [54], for five days instead of two days [40,50] (Appendix A).

Though, regarding *C. jejuni* and *C. coli,* comparison of isolation rates obtained by filtration culture and culture on different selective media, i.e., Skirrow, Butzler, modified Charcoal-Cefoperazone-Deoxycholate agar (mCCDA), and Cefoperazone-Amphotericin B-Teicoplanin (CAT) media, showed non-significant and contradictory differences depending on the filters and selective media used [32,40,49,52,56,60,61,64,66]. Consequently, filtration culture has never been recommended as a substitute for, but rather in complement to, culture on selective media, notably for paediatric or immunosuppressed patients, or in the case of an outbreak with no identified pathogen [32,46,49,60,64,66,67]. Moreover, filtration is labour-intensive and lengthy. In addition, the pathogenicity of the most common fastidious non-*jejuni*/*coli Campylobacter* sp., i.e., *C. concisus* [68,69,70,71], *C. curvus* [65], and *C. ureolyticus* [72,73,74], remains to be clarified [75,76,77,78,79]. This is particularly controversial when one considers, for example, that *C. concisus* has been detected in the saliva of healthy carriers [80] or that *C. concisus* [47] and *C. ureolyticus* [81] are found in statistically similar proportions in the stools of diarrhoeal patients and healthy individuals. Furthermore, *C. upsaliensis* infections, whose clinical relevance is better established [82,83,84,85], appear to be far less common than those caused by *C. jejuni* and *C. coli* [86]. As a result, selective media were introduced into routine microbiology in the 1980s [87], eventually leading to the recognition of campylobacteriosis as the leading bacterial cause of human gastroenteritis worldwide [75,88,89] and the most reported zoonosis in Europe since 2005 [86], with over more than twice as many reported cases as salmonellosis [86,90]. In fact, in the European Union (EU) and the United States, *Campylobacter* sp. infections are reported to be responsible for more than 10,000 and nearly 2000 hospitalisations per year, respectively [86,90,91,92]. In both regions, this has resulted in about 30 deaths per year [86,90,91,92]. However, filtration culture remains limited to a few reference laboratories, mostly for clinical epidemiological research rather than routine diagnosis [55,61,63,64,66]. Consequently, thermophilic *C. jejuni* and *C. coli* account for over 95% of reported cases [86], while most cases of acute gastroenteritis are of unknown aetiology [93,94], which is acceptable because these infections tend to be self-limiting. Additionally, when gastroenteritis is documented, viral causes are more common than bacterial causes [95].

In contrast, the genus *Campylobacter* currently comprises 45 species and 13 subspecies [96], of which more than 10 are associated with human infections [68,75,79]. Recently, non-*jejuni*/*coli Campylobacter* species have been associated with inflammatory bowel disease (IBD). In 2009, Zhang et al. [97] noted higher PCR detection of *C. concisus* in intestinal biopsies from children with Crohn’s disease than in controls (51% vs. 2%; *p* < 0.0001). Again, in 2010, Man et al. [98] reported a higher prevalence of PCR-detected *C. concisus* in stools of newly diagnosed Crohn’s disease patients than in healthy and non-inflammatory bowel disease controls (65% vs. 33% vs. 37%, respectively; *p* < 0.05). Then, in 2011, Mukhopadhya and colleagues [99] showed significantly higher prevalence rates of *C. concisus* and *C. ureolyticus* in colon biopsy specimens from adults with ulcerative colitis than in healthy controls (*p* < 0.01). Finally, in 2012, the *C. concisus* incidence rate among 8302 patients presenting gastroenteritis in North Jutland, Denmark, was measured at 35/100,000 inhabitants, almost as high as the *C. jejuni* plus *C. coli* incidence rate [100].

However, they are still seriously underdiagnosed [51,54,61,66,75,77], though with the development of genomics over the last two decades, the diversity of *Campylobacter* sp. has re-emerged [101]. In addition, rapid supplanting of culture-based methods in routine diagnostics by commercial PCR assays, also called culture-independent diagnostic tests, for the detection of mainly *C. jejuni*, *C. coli*, and, to a lesser extent, *C. upsaliensis* [102,103], is ongoing. As a result, funding for the culture of fastidious *Campylobacter* species is more threatened than ever.

Finally, a 28.1% global decrease in the annual notification rate of campylobacteriosis was observed among the 27 EU Member States when comparing the 2017–2019 period to 2021 [90]. Notably, the number of travel-associated cases was significantly lowered [90]. Indeed, in 6 out of 27 countries, including Belgium, a statistically significant decrease in the number of confirmed cases was observed when comparing the pre-COVID-19 period to 2021. This contrasts with the statistically significant increasing trend observed over the preceding seven-year period of 2008–2014 [104]. This decline was particularly evident in 2020, which was severely affected by the pandemic, resulting in a reduction in international travel [86,90,105]. In fact, this decreasing tendency between 2021 and the pre-COVID-19 period was observed for all zoonosis in Europe, with the exception of tularaemia [90]. In 2022, the notification rate in most EU Member States did not match that of recent pre-pandemic years [86].

In contrast, in 2020, Kuhn et al. [106] predicted a 25 and 196% increase in campylobacteriosis incidences by the end of the 2040s and the 2080s, respectively, in the four Nordic countries (i.e., Finland, Sweden, Norway, and Denmark), secondary to global warming and increases in precipitation, particularly heavy rainfall.

Considering all the upheavals in the clinical approach to Campylobacterales, this longitudinal clinical study aimed to evaluate their diversity among clinical samples, as well as the ongoing epidemiological dynamics in Brussels, Belgium, using an optimised filtration culture in parallel with selective inhibitory medium culture of stools collected for routine diagnosis.

## 2. Materials and Methods

### 2.1. Stool Samples

Over a 6-year period, from 2017 to 2019 (pre-COVID-19 period) and 2021 to 2023 (post-COVID-19 period), a total of 51,065 stool samples sent from 4 university hospitals in Brussels, Belgium, to the Laboratoire Hospitalier Universitaire de Bruxelles-Brussel Universitair Laboratorium (LHUB-ULB) for testing for bacterial gastrointestinal pathogens, were considered for analysis. Samples collected in 2020 were excluded because some culture reagents, in particular filters, were not available in the LHUB-ULB for more than six months due to the COVID-19 pandemic. Samples obtained from a fifth partner university hospital were excluded, as a distinct culture methodology was employed in the years 2017 and 2018.

Strains belonging to the same species exhibiting the same antibiotic susceptibility profile, isolated from stools collected within less than a one-month interval from the same patient, were considered as related to the same clinical episode. Only the first collected stool from each episode was included in the study, with the following stools being considered as duplicates and excluded from positive stool statistics.

### 2.2. Culture

Upon arrival at the LHUB-ULB, the samples were stored at 4 °C. They were inoculated into media within 24 h of their arrival, with the exception of samples received on Friday after 3 pm and on Saturday, which were inoculated on the following Monday (within 72 h of their arrival). For Campylobacterales isolation, samples were plated on a selective inhibitory agar medium (Campylobacter Selective Agar (Butzler)™, ThermoFisher Scientific, Waltham, MA, USA) and incubated for 48 h at 40–43 °C in a microaerobic atmosphere (85% N_2_—10% CO_2_—5% O_2_—0% H_2_). In parallel, 2 mL of liquid stool (or liquefied soft stool by the addition of isotonic saline) was diluted 1:3 in a Brucella broth (EO Labs, Cumbernauld, UK) and incubated for 30 min at 35–38 °C in the same atmosphere as used in the same atmosphere as used for Butzler plates Butzler plates (85% N_2_—10% CO_2_—5% O_2_—0% H_2_). Six drops of this broth were then transferred to the surface of a 0.6 µm polycarbonate filter (Whatman™ Nuclepore™ Hydrophilic Membrane, Cytiva, Marlborough, MA, USA), placed on an antibiotic-free Columbia agar containing 5% sheep’s blood (BD™), and incubated at 35–38 °C in the same atmosphere again (85% N_2_—10% CO_2_—5% O_2_—0% H_2_). The Brucella broth was re-incubated for a further 30 min. Then, a further six drops of the re-incubated broth were transferred to the surface of the same filter and the Columbia agar was re-incubated for a further 30 min. Finally, the filters were removed, and the Columbia agars were incubated for five days at 35–38 °C in a microaerobic atmosphere enriched with H_2_ (80% N_2_—7% CO_2_—6% O_2_—7% H_2_).

### 2.3. Identification

Butzler plates were examined after 24 and 48 h of incubation, while Columbia plates were examined after 48 h and 5 days of incubation. All growing colonies were identified using matrix-assisted laser desorption ionisation–time of flight mass spectrometry (MALDI-TOF MS; Biotyper database; Bruker Daltonics, Bremen, Germany). The databases utilized were those that have been employed over time. Each new iteration was validated using a diverse panel of *Campylobacter* species, in accordance with our quality assurance protocols.

### 2.4. Statistical Analyses

The proportions of Campylobacter isolates from the pre-COVID-19 (2017–2019) and post-COVID-19 (2021–2023) pandemic periods were compared using the Chi-squared test, and continuous data were compared using the non-parametric Mann–Whitney test. A *p*-value of <0.05 was considered statistically significant.

## 3. Results

### 3.1. Prevalence and Species Repartition

The mean positivity rate of stool cultures for Campylobacterales over the entire study period was 7.1%, with annual positivity rates ranging from minimum 5.9% (2018) to maximum 8.0% (2021). Sixteen species of the genus *Campylobacter* and four species of related organisms were identified. Throughout the study period, *C. concisus* and *C. curvus* represented 36.4–51.2% and 4.3–9.8% (minimum–maximum), respectively, of annual isolates, totalling 1600 and 272 strains, whereas *C. jejuni* and *C. coli* represented only 20.9–32.2% and 1.9–5.0% (minimum–maximum), respectively, of annual isolates, totalling 970 and 112 strains (Table 1). Interestingly, 447 *C. ureolyticus* strains were isolated, representing 3.7–23.0% of the annual isolates (minimum in 2018—maximum in 2023, respectively). It was the fourth most common species after *C. coli* in the pre-COVID-19 period, but the third most common in the post-COVID-19 period, with almost the same incidence as *C. jejuni* infections. *C. upsaliensis* and *Aliarcobacter butzleri* were the next most common species isolated, averaging 1.5 and 1.4% annually (*n* = 53 and 50 strains, respectively; Table 1).

Another thirteen species combined accounted for 2.8% of the total. Among them, 45 strains belonging to 10 additional *Campylobacter* species were identified, i.e., *C. showae* (*n* = 25), *C. hyointestinalis* (*n* = 7), *C. fetus* (*n* = 5), *C. gracilis* (*n* = 2), *C. lari* (*n* = 2), *C. hominis* (*n* = 1), *C. rectus* (*n* = 1), *C. sputorum* (*n* = 1), *C. lanienae* (*n* = 1), and *C. peloridis* (*n* = 1)*,* accounting in total for only 1.3% of all isolates. In addition to *Campylobacter* sp. isolates, 44 strains from 2 *Helicobacter* species were identified, namely, *H. pullorum* (*n* = 33) and *H. cinaedi* (*n* = 11), representing 1.2% of all isolates. Finally, in addition to *A. butzleri*, 10 *Aliarcobacter cryoaerophilus* strains were isolated (Table 1).

### 3.2. Dynamics from 2017 to 2023

There were 28,911 and 22,154 stool samples collected during the pre- and post-COVID-19 periods, respectively, with mean positivity rates of 6.7 and 7.5%, respectively.

Figure 1 shows the trends in the number of isolates of the six most common *Campyloacter* sp. collected throughout the study period.

In contrast between these two time periods, the mean numbers of *C. jejuni, C. coli*, and *C. concisus* isolates decreased by 37, 53, and 28%, respectively (*p* < 0.05), while the mean number of *C. curvus* isolates increased by 18%, and those of *C. upsaliensis* isolates remained stable. Mean numbers of *Aliarcobacter* sp. and *Helicobacter* sp. also decreased after the COVID-19 pandemic by 24 and 57%, respectively.

Finally, *C. ureolyticus*, which accounted for only 3.5, 4.0, and 9.0% of the yearly isolates in 2017, 2018, and 2019, respectively, accounted for 20.1, 18.7, and 23.0% of the yearly isolates in 2021, 2022, and 2023 (*p* < 0.05; Table 1).

### 3.3. Seasonal Prevalence

Non-uniform distributions of the incidence of *C. jejuni* and *C. ureolyticus* were found across the 12 months of the year (*p* < 0.05; Figure 2a,b), while the incidence of *C. concisus* and *C. curvus* showed no seasonal trend (Figure 2c).

*C. jejuni* isolates peaked twice each year, in January (99 isolates) and in July (113 isolates)/August (115 isolates), representing 33.7% of the total (970 isolates; Figure 2a).

*C. ureolyticus* incidence increased from January to April (51.9% of the total number of isolates), and decreased from May to December, with the fewest *C. ureolyticus* detections in November (Figure 2b).

### 3.4. Age Distribution

With regard to the age profile, a unimodal distribution was observed for *C. jejuni* isolates, 56% of which were detected in children under the age of 10 years. Conversely, *C. curvus* and *C. ureolyticus* were bimodally distributed, with 13 and 23% detected in children under 10 years and 42 and 25% isolated from patients aged 50 to 69 years (Figure 3).

## 4. Discussion

### 4.1. Prevalence and Species Repartition

In line with the literature on stool filtration culture, the species distribution among *Campylobacter* sp.-positive samples reflected a much greater diversity than that commonly described by the vast majority of routine diagnostic laboratories [86,87,105] using either standard culture on selective inhibitory medium or commercial multiplex PCR assays (e.g., BD Max, Seegene, Qiastat, or FilmArray assays), which only allow the detection of *C. jejuni, C. coli*, and rarely, *C. upsaliensis* and *C. fetus*.

Firstly, *C. concisus* was the most common isolated species in our laboratory (Table 1). As it was twice as frequent as *C. jejuni*, it seems to be the most common Campylobacter species encountered in the gastrointestinal tract.

These findings agree with the literature. Indeed, when the Cape Town protocol [50] (0.6 μm pore cellulose acetate filters and tryptose agar plates containing 10% un-lysed sheep or horse blood) was employed as an alternative to yeast-enriched blood agar plates incubated in a H₂-enriched microaerobic atmosphere, Lastovica et al. [55] discovered that 24.63% of the 5443 Campylobacterales strains isolated from a paediatric population were *C. concisus* (Appendix A). Subsequently, Nielsen et al. [60] demonstrated that polycarbonate is a more efficient stool filtration material than cellulose acetate using 0.6 μm pore size filters: 26% more *C. concisus* strains was observed in 1791 diarrhoeal stools collected in Denmark in 2012 after replacing cellulose acetate with polycarbonate (114/134 vs. 79/134; *p* < 0.0001) (Appendix A). The authors suggested that the smooth, glassy surface of the polycarbonate filter would be more suitable for the penetration of motile *Campylobacter* sp. than the rough surface of the cellulose acetate filter, which could catch random particles and block *Campylobacter* sp. from passing through the filter [60].

In contrast, only 1.9% of 1394 stool isolates were *C. concisus* (*n* = 27) in a study by Vandenberg et al. [56] in Belgium, who adapted the Cape Town protocol, replacing tryptose agar plates containing 10% un-lysed blood with Mueller–Hinton agar plates with 5% sheep blood agar, and 0.6 μm with 0.45 μm pore size cellulose filters (Appendix A). This could be explained by the fact that the 0.45 µm pore size is too small for motile *Campylobacter* sp., especially *C. concisus*, to efficiently penetrate the filter.

The superior efficiency of the polycarbonate filter with a 0.6 μm pore size over the cellulose filter with 0.45 μm pore size for the isolation of non-*jejuni/coli Campylobacter* sp. was again confirmed by Nachamkin et al. in 2017 [63] and by our team in 2019 [64] (Appendix A). In addition, we showed that *C. concisus* isolation was also facilitated by the use of Columbia agar instead of blood-enriched Mueller–Hinton agar incubated in an H_2_-enriched atmosphere. With regard to *C. curvus* and *C. ureolyticus*, although studies [40,51,52,55,56,61,63,64] include small numbers of isolates, the use of 0.45 μm pore size filters also appears to be a dramatic limiting factor on their growth (Appendix A). Indeed, in our laboratory, when we switched from 0.6 μm pore size polycarbonate filters, which were out of stock due to the COVID-19 pandemic, to 0.45 μm pore size cellulose acetate filters (Porafil CA, Macherey-Nagel) from March to September 2020, we observed a more than 90% decrease in *C. concisus*, *C. ureolyticus*, and *C. curvus* isolation rates.

Only 53 *C. upsaliensis* strains were isolated from 51,065 stool samples, resulting in a prevalence of 0.1% (Table 1). These results are consistent with those reported in Denmark in 2000 by Engberg et al. [40]. Using three selective media and a filtration technique, they were unable to isolate *C. upsaliensis* from 1376 clinical stool samples from patients with diarrhoea (Appendix A). Also in Denmark, Nielsen et al. [60] isolated only 1 *C. upsaliensis* strain from 1791 diarrhoeal stools (prevalence: 0.06%) collected in 2012 using the same technique, and 2 years later, cultures yielded only 5 *C. upsaliensis* strains from 5963 diarrhoeal stools (prevalence: 0.08%) [61] (Appendix A). The same observation was made by Nachamkin et al. [63] in Pennsylvania, US, who isolated 7 distinct *Campylobacter* species, but not *C. upsaliensis*, from 225 faecal samples collected in 2016. This was despite performing filtration culture on three types of filters: cellulose acetate filters of 0.45 and 0.65 μm pore sizes, and polycarbonate filters of 0.6 μm pore size, as well as Brucella agar plates containing 5% sheep blood, hemin, and vitamin K (Appendix A). More recently, our team [64] attempted to optimize the filtration technique by evaluating the influence of the agar (Mueller–Hinton versus Columbia agar, both containing 5% sheep blood), the filter (0.6 μm polycarbonate versus 0.45 μm cellulose acetate), and the atmosphere (7% H_2_-enriched versus non-H_2_-enriched microaerobic atmosphere) on more than 2000 stool samples collected in Brussels, Belgium, in 2014, 2016, and 2018. Irrespective of the parameters, less than one in five hundred stool cultures yielded *C. upsaliensis* (prevalence <0.2%) (Appendix A). This is again in line with a previous study carried out in our laboratory in 2004, where only 85 *C. upsaliensis* were isolated from 67,599 stools (prevalence: 0.13%) collected between 1995 and 2002, performing filtration culture with Mueller–Hinton agar containing 5% sheep blood and a 0.45 μm pore size cellulose acetate filter [56] (Appendix A).

However, significantly higher isolation rates of *C. upsaliensis* from stool samples were reported by Goossens et al. in Belgium in 1989 (prevalence: 0.73%) [46] and Lindblom et al. in Sweden in 1993 (prevalence: 0.82%) [84]. Furthermore, in 2000, the Cape Town protocol allowed 4122 strains belonging to *Campylobacter* and related species to be isolated from 19,535 stool samples collected from children with diarrhoea in Cape Town, South Africa, between 1990 and 2000, of which 23% were identified as *C. upsaliensis* (prevalence: 4.9%) [51].

It can be hypothesized that the choice of an appropriate growth medium, e.g., tryptose agar containing 10% un-lysed sheep or horse blood, as recommended by the Cape Town protocol, is an overriding constraint for the culture of *C. upsaliensis*. Therefore, the prevalence of *C. upsaliensis* in Belgium may be higher than that reported here. Further studies comparing agar media among themselves or with results obtained by molecular techniques are needed to clarify this point. However, local animal-to-human and human-to-human transmission, especially among young children, travellers, and immigrants, may contribute to the higher prevalence observed in some cohorts in some world regions compared to Belgium [107].

As far as *Aliarcobacter* sp. are concerned, *A. butzleri* and *A. cryoaerophilus* have persistently represented less than 1.5% of isolates per year (Table 1). Conversely, Vandenberg et al. [56], in the same laboratory, reported that 4.0% of the 1906 isolates obtained from 67,599 stools collected between 1995 and 2002 belonged to the genus *Aliarcobacter*. Nevertheless, 71 of the 77 strains were isolated only by the method of De Boer et al. [108], which consists of 24 h selective enrichment of 0.5 g of stool in Brucella broth supplemented with antibiotics, followed by culture on *Aliarcobacter* selective plates incubated for 3 days at 25 °C in a microaerobic atmosphere. Specific conditions for culturing *Aliarcobacter* sp. were also recommended by Lastovica and Le Roux [50], according to the Cape Town protocol. By not following these recommendations, Lastovica et al. failed to isolate *A. cryoaerophilus* strains from 19,535 (2000) and more than 20,000 (2006) diarrhoeal stools, and *A. butzleri* isolates represented only 0.39 and 0.36% of all strains grown in these studies [51,55]. As a result, the prevalence of *Aliarcobacter* sp. in clinical stools reported here may be underestimated. It also confirms the need for using a culture method suitable for *Aliarcobacter* sp. that includes incubation below 35 °C.

### 4.2. Dynamics from 2017 to 2023

The mean number of *C. ureolyticus* annual isolates increased by 218% (from 36 to 113; *p* < 0.05; Table 1) between the pre- and post-COVID-19 periods, even exceeding the *C. jejuni* rate in 2023 (Figure 1). At the same time, the mean annual number of *C. jejuni*, *C. coli*, and *C. concisus* isolates decreased by 37, 53, and 28% (from 198 to 125, 25 to 12, and 310 to 223, respectively; *p* < 0.05; Table 1). Implementing hygienic measures in 2020 may also have accelerated an existing trend. However, it is interesting to note that the annual isolation rate of *C. ureolyticus* had already doubled between 2018 and 2019 (from 23 to 57, respectively; Table 1), before the COVID-19 pandemic. As the variations are in opposite directions, they cannot be explained by a global decrease or increase in laboratory activity, even though 28,911 stools were cultured in the pre-COVID-19 period, while only 22,154 were analysed in the post-COVID-19 years.

Such a high proportion of *C. ureolyticus* among stool-detected *Campylobacter* sp. was previously reported by Bullman et al. in 2011 [109], before the COVID-19 pandemic, conducting in-house genus- and species-specific PCR testing of 7194 samples. In this study, 27.3% of the 373 *Campylobacter* sp. detected in the stools of 349 diarrhoeal patients in Southern Ireland were non-*jejuni*/*coli Campylobacter* species, of which 81.7% were actually *C. ureolyticus* (*n* = 83 stools). In addition, Hatanaka et al. [110] reported that 51.9% of *Campylobacter* sp. detected by PCR in the stools of children with diarrhoea in Japan were *C. ureolyticus*. Conversely, using PCR, Collado et al. [78] detected *C. ureolyticus* in low and statistically similar proportions in stools from both diarrhoeal and healthy groups.

To confirm the increase in the presence of *C. ureolyticus* in human clinical samples and explain the possible partial replacement of *C. jejuni*, *C. coli*, and *C. concisus* with *C. ureolyticus*, further studies are needed, especially on stools collected in other regions of the world. Indeed, several authors have predicted an increase in the incidence of campylobacteriosis in Europe and Asia as a consequence of climate change, together with an increase in temperature, humidity, and especially, heavy precipitation [111,112,113]. However, these predictions may not apply to *C. ureolyticus*, as very little is known about its reservoir and transmission route to humans [114,115]. In addition, studies over a longer period of time, together with climate observations, are needed in order to make further assumptions.

### 4.3. Seasonality

In line with the literature, *C. jejuni* incidence peaked in July/August [86,90,109], whereas *C. ureolyticus* incidence increased from January to April [109] (coinciding with increased cattle shedding) [72,73,74] (Figure 2). Notably, *C. ureolyticus* has been detected by PCR in unpasteurised milk in Ireland [114] and in cat, cow, and pig faeces [115]. This is consistent with studies suggesting that *C. ureolyticus* is not human commensal [72,74]. Furthermore, regarding *Campylobacter* sp. bacteraemia, Tinévez et al. [116] reported that in a nationwide study conducted in France from 2015 to 2019, 22 out of 592 episodes (3.7%) were attributed to *C. ureolyticus*, which was the fourth most frequently isolated species after *C. jejuni*, *C. fetus*, and *C. coli* (42.9, 42.6, and 6.8%, respectively). This also indicates the potential invasiveness of this species, for which an increasing number of putative virulence factors have been described [72,73,74]. Moreover, when *C. ureolyticus* was isolated, it was observed that in over 95% of cases, no additional Campylobacterales were present, thus supporting its involvement in the observed symptoms.

Besides, the incidence of *C. concisus* showed no seasonal trend (Figure 2), as observed in Denmark by Nielsen et al. [100]. This tends to indicate that humans commensally carry this species, in agreement with Zhang et al. [80] and Macuch et al. [117], who suggested that the human oral cavity may serve as a reservoir for this species. Consistent with this hypothesis, neither Tinévez et al. [116], as the French Campylobacter NRC from 2015 to 2019, Lastovica et al. [55] in South Africa from 1990 to 2005, nor our laboratory, as the Belgium Campylobacter NRC from 2014 to 2023 [118], reported *C. concisus* bacteraemia. In addition, only two *C. curvus* blood isolates were obtained in France. This does not support the invasiveness of these two species, especially in view of the important detection rate of *C. concisus* in human stools. The possibility exists that cases of *C. concisus* and *C. curvus* bacteraemia remain undiagnosed owing to the inability of the bacteria to grow in a commercial blood culture bottle, in particular due to a lack of H_2_ enrichment of the atmosphere. However, the susceptibility of *C. concisus* to the bactericidal effects of human serum has been clearly demonstrated. Consequently, this bacterium is likely to be less able to cause bacteraemia [119].

### 4.4. Age Distribution

Our results (Figure 3) agree with those of Bullman et al. [109], with *C. ureolyticus* and *C. curvus* being more prevalent at the two extremes of life when immunity is compromised, in contrast to *C. jejuni*, whose cases were mostly reported in children. This suggests that *C. ureolyticus* is more of an opportunistic pathogen. However, three putative virulence and colonisation factors, the surface antigen CjaA, an outer membrane fibronectin-binding protein, and an S-layer RTX toxin, have been detected in its secretome [72].

## 5. Conclusions

In conclusion, 3604 strains belonging to 20 species of *Campylobacter* and related organisms were isolated from 51,065 stool samples collected over 6 years (2017–2019 and 2021–2023, pre- and post-COVID-19 periods, respectively) using filtration culture. The most frequently detected species was *C. concisus*, which accounted for almost half of all isolates. Meanwhile, the annual isolation rate of *C. ureolyticus* increased by 218% between pre- and post-COVID-19 periods. This raises concerns about the pathogenic potential of these species, particularly in vulnerable patients at risk of non-self-limiting infections. Currently, a growing number of laboratories are diagnosing campylobacteriosis using commercial multiplex PCR assays, the efficacy of which remains to be evaluated in the context of such emerging species. Furthermore, these assays provide no information about antimicrobial resistance. It is, therefore, recommended that a routine reflex culture be carried out in the event of a positive multiplex PCR result. In the event of a negative result in a patient exhibiting persistent gastrointestinal symptoms with an unidentified aetiology, a control stool sample may also be collected and submitted to an NRC that performs filtration culture.

## Figures and Tables

**Figure 1 pathogens-13-00475-f001:**
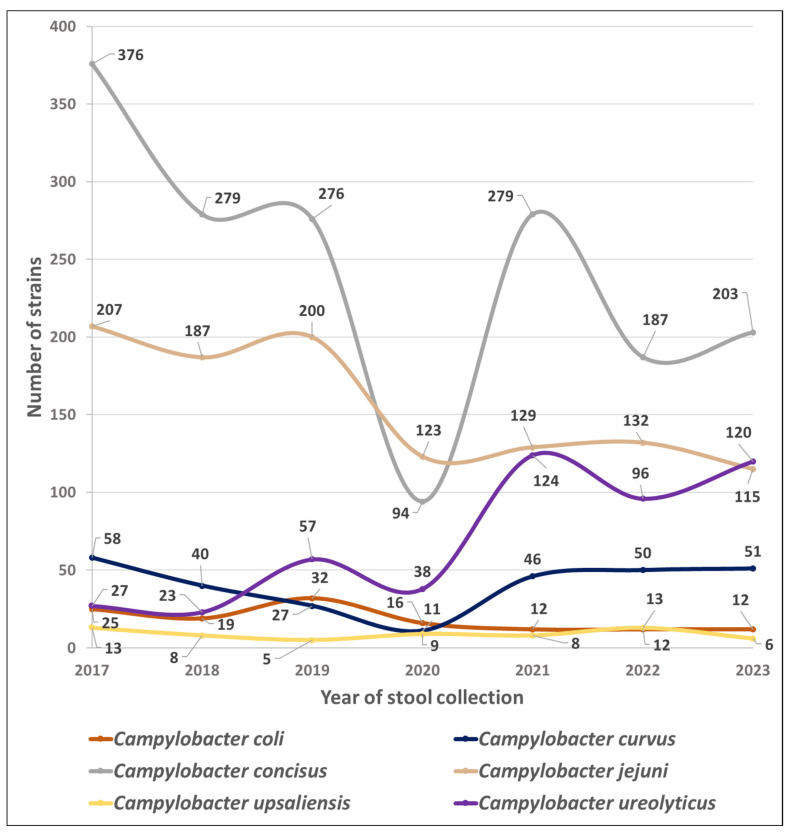
Species prevalence dynamics of Campylobacterales in 51,065 stools at the Department of Microbiology, LHUB-ULB, Belgium, from 1 January 2017 to 31 December 2023, using Butzler selective medium plus filtration culture with a 0.6 µm pore size polycarbonate filter and Columbia agar. Percentages indicate the proportion of each species in the total number of strains of Campylobacterales in a year.

**Figure 2 pathogens-13-00475-f002:**
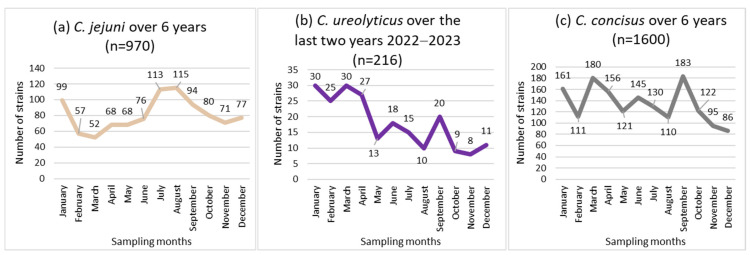
Seasonal distribution of the most prevalent *Campylobacter* species, i.e., *C. jejuni* (**a**), *C. concicus* (**b**), and *C. ureolyticus* (**c**) in 51,065 stools at the Department of Microbiology, LHUB-ULB, Belgium, from 1 January 2017 to 31 December 2023, excluding year 2020 using Butzler selective medium plus filtration culture, with a 0.6 µm pore size polycarbonate filter and Columbia agar.

**Figure 3 pathogens-13-00475-f003:**
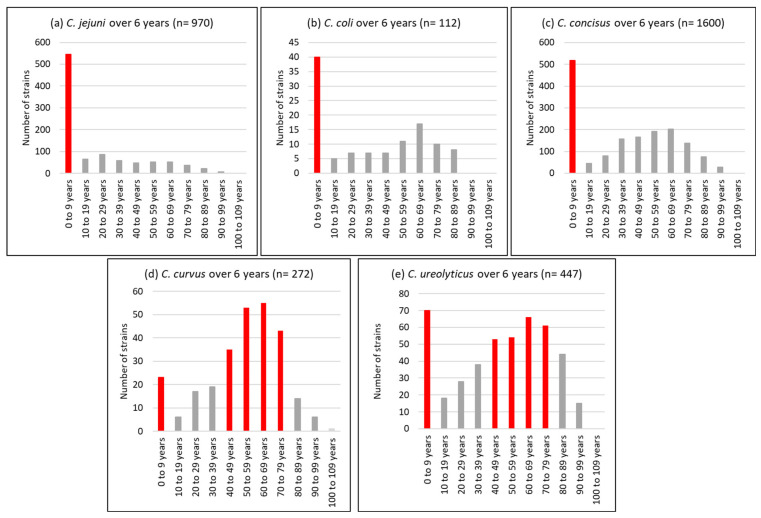
Age profile of the most prevalent *Campylobacter* species, i.e., *C. jejuni* (**a**), *C. coli* (**b**), *C. concisus* (**c**), *C. curvus* (**d**), and *C. ureolyticus* (**e**) in 51,065 stools at the Department of Microbiology, LHUB-ULB, Belgium, from 1 January 2017 to 31 December 2023, excluding year 2020 using Butzler selective medium plus filtration culture, with a 0.6 µm pore size polycarbonate filter and Columbia agar.

**Table 1 pathogens-13-00475-t001:** Species prevalence of Campylobacterales in 51,065 stools at the Department of Microbiology, LHUB-ULB, Belgium, from 1 January 2017 to 31 December 2023, using Butzler selective medium plus filtration culture with a 0.6 µm pore size polycarbonate filter and Columbia agar. Percentages indicate the proportion of each species in the total number of strains of Campylobacterales in a year.

*Campylobacter* and Related Species	2017	% in 2017	2018	% in 2018	2019	% in 2019	2021	% in 2021	2022	% in 2022	2023	% in 2023	Total over 6 Years	Mean % before COVID-19	Mean % after COVID-19	Mean % over 6 Years
*C. concisus*	376	51.2	279	47.9	276	43.5	279	45.1	187	36.4	203	38.9	1.6	47.7	40.4	44.4
*C. jejuni*	207	28.2	187	32.1	200	31.5	129	20.9	132	25.7	115	22.0	970	30.5	22.7	26.9
*C. ureolyticus*	27	3.7	23	4.0	57	9.0	124	20.1	96	18.7	120	23.0	447	5.5	20.6	12.4
*C. curvus*	58	7.9	40	6.9	27	4.3	46	7.4	50	9.7	51	9.8	272	6.4	8.9	7.5
*C. coli*	25	3.4	19	3.3	32	5.0	12	1.9	12	2.3	12	2.3	112	3.9	2.2	3.1
*C. upsaliensis*	13	1.8	8	1.4	5	0.8	8	1.3	13	2.5	6	1.1	53	1.3	1.6	1.5
*C. showae*	5	0.7	3	0.5	2	0.3	5	0.8	6	1.2	4	0.8	25	0.5	0.9	0.7
*C. hyointestinalis*	0	0	1	0.2	1	0.2	0	0	2	0.4	3	0.6	7	0.1	0.3	0.2
*C. fetus*	0	0	3	0.5	1	0.2	0	0	0	0	1	0.2	5	0.2	0.1	0.1
*C. gracilis*	0	0	0	0	2	0.3	0	0	0	0	0	0	2	0.1	0	0.1
*C. lari*	0	0	1	0.2	1	0.2	0	0	0	0	0	0	2	0.1	0	0.1
*C. hominis*	0	0	0	0	1	0.2	0	0	0	0	0	0	1	0.1	0	0.03
*C. rectus*	1	0.1	0	0	0	0	0	0	0	0	0	0	1	0.1	0	0.03
*C. sputorum*	1	0.1	0	0	0	0	0	0	0	0	0	0	1	0.1	0	0.03
*C. lanienae*	0	0	1	0.2	0	0	0	0	0	0	0	0	1	0.1	0	0.03
*C. peloridis*	0	0	0	0	0	0	1	0.2	0	0	0	0	1	0	0.1	0.03
*A. butzleri*	9	1.2	10	1.7	16	2.5	7	1.1	6	1.2	2	0.4	50	1.8	0.9	1.4
*A. cryoaerophilus*	1	0.1	1	0.2	5	0.8	2	0.3	1	0.2	0	0	10	0.4	0.2	0.3
*H. cinaedi*	3	0.4	0	0	2	0.3	2	0.3	2	0.4	2	0.4	11	0.3	0.4	0.3
*H. pullorum*	8	1.1	6	1.0	6	0.9	3	0.5	7	1.4	3	0.6	33	1.0	0.8	0.9
Total per year	734		582		634		618		514		522		3.604			
Postivity rate	7.3		5.9		7.0		8.0		6.6		7.8		7.1			
Stools cultured per year	10.094		9.793		9.024		7.717		7.778		6.659		51.065			

## Data Availability

The original contributions presented in the study are included in the article, further inquiries can be directed to the corresponding author.

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
