# Peer review of "Assessing the Prevalence and Dynamics of Emerging Campylobacterales in Human Stool Samples in Brussels by Filtration Culture"

_pathogens, 2024, doi:10.3390/pathogens13060475_

Round 1

Reviewer 1 Report

Comments and Suggestions for Authors

Manuscript title: Assessing the prevalence and dynamics of emerging Campylobacterales in human stool samples in Brussels by filtration culture.

Manuscript ID/ File Name: pathogens-3017969-peer-review-v1

Authors: Giraudon et al.

Campylobacters are indeed an important enteric pathogen for humans with a global footprint. It is true that these pathogens are a major concern in many continents including Europe, Asia, and Americas. Despite progress in assuring hygienic food supply chain and general environmental sanitary measures, they continue to emerge. Often new member species of the group are attributed for human illnesses. Clinical management of infection by campylobacters relies on antimicrobial therapy, which is also being compromised due to rising antimicrobial resistance among these organisms. In this context, the present study is interesting, and I enjoyed reading the manuscript.

In this study, Giraudon et al. analyzed the prevalence of organisms belonging to the order Campylobacterales in human faecal samples. The authors analyzed 51065 stool samples over 6 years period that was split into pre-COVID and post-COVID years. Authors’ approach to isolation and identification of the organisms involved selective culture and filtration technique followed by confirmation through MALDI-TOF. The positivity data recorded by the authors, were then analyzed to infer trends in occurrences.

Despite the relevance of the study there are certain weaknesses in the study that need to be addressed by the authors before being published.

Major concern(s):

1.      The attempt by the authors to link the increased occurrence of some species of campylobacters with the change in climate conditions appears to be too conjectural. This is because authors did not present any data on the climatic zones in which the patients lived, nor did they attempt to collect and present any data on climate conditions. In fact, 6 years would be rather a short span of time to infer climatic trends.

2.      While COVID-19 pandemic caused global disruption affecting almost all facets of life throughout the world, relating the change in occurrence of certain campylobacter species, requires more convincing data, else it remains anecdotal. I believe that without these two issues, the manuscript is still strong enough to stand on its own.

3.      At some point, the manuscript appears too loaded and too leading.

I suggest that the authors reduce the length of the manuscript, especially the discussion section and the manuscript would appear crisper, as the manuscript contains important long-term data.

Introduction:

1.      Line 111-112: Please rephrase. In fact, lines 106-115 require improvement in clarity.

Materials and Methods:

1.      Line(s) 122: Please mention sample size in this section.

2.      Line(s) 144-145 and 147-148: Please clarify ‘same atmosphere’.

3.      Line(s) 157-158: Please mention MALDI make and firm name. Also mention database supplier name and version.

Results:

1.      Table 1: Please correct ‘COVID-20’!.

2.      Line(s) 77-78: Sentence not clear. Please rephrase.

Discussion:

1.      Please reduce the discussion section by approximately 50%.

2.      Line(s) 419-421: These assertions require suitable references.

3.      A short paragraph concluding the noteworthy results is needed.

Comments on the Quality of English Language

 Minor editing of English language required

Author Response

Dear Editor,
We would like to thank you and the reviewers for your time and careful review of our work. We edited our manuscript following your recommendations. Responses to the reviewers are displayed below.
As requested, we provide enclosed the revised manuscript in two versions: one clean revised version and a version with track changes so that you can see where the changes have been made.

1) Reviewer 1
Major concerns:
1. The attempt by the authors to link the increased occurrence of some species of campylobacters with the change in climate conditions appears to be too conjectural. This is because authors did not present any data on the climatic zones in which the patients lived, nor did they attempt to collect and present any data on climate conditions. In fact, 6 years would be rather a short span of time to infer climatic trends.

The manuscript has undergone the requisite revisions. It was noted that further studies, conducted in other regions of the world and over longer period of time are needed to confirm an increase in the occurrence of some Campylobacter species and to further investigate the relationship between such an increase and climate change.

2. While COVID-19 pandemic caused global disruption affecting almost all facets of life throughout the world, relating the change in occurrence of certain campylobacter species, requires more convincing data, else it remains anecdotal. I believe that without these two issues, the manuscript is still strong enough to stand on its own.

The manuscript has now been revised to include the clarification that the hypothesis regarding the acceleration of the decline in cases of C. jejuni and C. coli infections following the implementation of hygiene measures in 2020 is still only that, namely, a hypothesis.

3. At some point, the manuscript appears too loaded and too leading.

The manuscript, and in particular the discussion, has been shortened in accordance with your recommendation.

Introduction

  1. Line 111-112: Please rephrase. In fact, lines 106-115 require improvement in clarity. The entire paragraph has undergone a comprehensive rewrite, as follow: “Finally, a 28.1% global decrease in the annual notification rate of campylobacteriosis is observed among the 27 EU Member States when comparing the 2017-2019 period to 2021. Notably, the number of travel-associated cases is significantly lowered [90]. Indeed, in six out of 27 countries, including Belgium, a statistically significant decrease in the number of confirmed cases is observed when comparing the pre-COVID-19 period to 2021. This decline is particularly evident in 2020, which was severely affected by the pandemic, resulting in a reduction in international travel [86,90,103]. In fact, this decreasing tendency between 2021 and the pre-COVID-19 period was observed for all zoonosis in Europe, with the exception of tularemia [90]. In 2022, the notification rate in most EU Member States did not match that of recent pre-pandemic years [86].”Materials and Methods:1. Line(s) 122: Please mention sample size in this section.The revised manuscript mentions sample size in the Materials and Methods section.

2. Line(s) 144-145 and 147-148: Please clarify ‘same atmosphere’.

In the revised manuscript, this has been clarified, as follow: “In parallel, 2 mL of liquid stool (or liquefied soft stool by the addition of isotonic saline) was diluted 1:3 in a Brucella broth (EO Labs, UK) and incubated for 30 minutes at 35-38°C in the same atmosphere than Butzler plates (85% N2 - 10% CO2 - 5% O2 - 0% H2). Six drops of this broth were then transferred to the surface of a 0.6 μm polycarbonate filter (Whatman™ Nuclepore™ Hydrophilic Membrane, Cytiva) placed on an antibiotic-free Columbia agar containing 5% sheep's blood (BD™) and incubated at 35-38°C in the same atmosphere again (85% N2 -10% CO2 - 5% O2 - 0% H2).”

3. Line(s) 157-158: Please mention MALDI make and firm name. Also mention database supplier name and version.

This has been clarified in the revised manuscript. “Matrix-assisted laser desorption ionisation-time of flight mass spectrometry (MALDI-TOF MS; Biotyper database; Bruker Daltonics, Bremen, Germany) was used to identify all growing colonies. The databases used are those used over time. Each new iteration was validated against a diverse panel of Campylobacter species in accordance with our quality assurance protocols.”

Results:
1. Table 1: Please correct ‘COVID-20’!.

The revised manuscript mentions sample size in the Materials and Methods section.

2. Line(s) 144-145 and 147-148: Please clarify ‘same atmosphere’.

In the revised manuscript, this has been clarified, as follow: “In parallel, 2 mL of liquid stool (or liquefied soft stool by the addition of isotonic saline) was diluted 1:3 in a Brucella broth (EO Labs, UK) and incubated for 30 minutes at 35-38°C in the same atmosphere than Butzler plates (85% N2 - 10% CO2 - 5% O2 - 0% H2). Six drops of this broth were then transferred to the surface of a 0.6 μm polycarbonate filter (Whatman™ Nuclepore™ Hydrophilic Membrane, Cytiva) placed on an antibiotic-free Columbia agar containing 5% sheep's blood (BD™) and incubated at 35-38°C in the same atmosphere again (85% N2 -10% CO2 - 5% O2 - 0% H2).”

3. Line(s) 157-158: Please mention MALDI make and firm name. Also mention database supplier name and version.

This has been clarified in the revised manuscript. “Matrix-assisted laser desorption ionisation-time of flight mass spectrometry (MALDI-TOF MS; Biotyper database; Bruker Daltonics, Bremen, Germany) was used to identify all growing colonies. The databases used are those used over time. Each new iteration was validated against a diverse panel of Campylobacter species in accordance with our quality assurance protocols.”

Results:

  1. Table 1: Please correct ‘COVID-20’!.

Correction has been made.

2. Line(s) 77-78: Sentence not clear. Please rephrase.

I assume this comment is intended for lines 177-178. This sentence was rephrased for clarification: “Interestingly, 447 C. ureolyticus strains were isolated, representing 3.7-23.0% of the annual isolates (min in 2018 - max in 2023, respectively). It was the fourth most common species after C. coli in the pre-COVID-19 period, but the third most common in the post-COVID-19 period, with almost the same incidence as C. jejuni infections. C. upsaliensis and Aliarcobacter butzleri were the next commonest species isolated, averaging 1.5 and 1.4% annually (n= 53 and 50 strains, respectively) (Table 1).”

Discussion:

1. Please reduce the discussion section by approximately 50%.

The discussion in the submitted manuscript (236 lines) has been shortened by almost half (165 lines). It has been divided into several paragraphs for the sake of clarity.

2. Line(s) 419-421: These assertions require suitable references.

References have been added to these assertions: “According to the literature, C. jejuni incidence peaks in July/August [86,90,109], whereas C. ureolyticus incidence rises from January to April [109] (coinciding with increased cattle shedding) [74] (Figure 2). This is consistent with studies suggesting C. ureolyticus is not human commensal [72,74].”

References:
72. Burgos-Portugal JA, Kaakoush NO, Raftery MJ, Mitchell HM. Pathogenic potential of Campylobacter ureolyticus. Infect Immun. 2012 Feb;80(2):883-90. doi: 10.1128/IAI.06031-11. Epub 2011 Nov 28. PMID: 22124656; PMCID: PMC3264317.
74. Maki JJ, Howard M, Connelly S, Pettengill MA, Hardy DJ, Cameron A. Species Delineation and Comparative Genomics within the Campylobacter ureolyticus Complex. J Clin Microbiol. 2023 May 23;61(5):e0004623. doi: 10.1128/jcm.00046-23. Epub 2023 Apr 27. PMID: 37129508; PMCID: PMC10204631.
86. European Food Safety Authority (EFSA); European Centre for Disease Prevention and Control (ECDC). The European Union One Health 2022 Zoonoses Report. EFSA J. 2023 Dec 12;21(12):e8442. doi: 10.2903/j.efsa.2023.8442. PMID: 38089471; PMCID: PMC10714251.
90. European Food Safety Authority; European Centre for Disease Prevention and Control. The European Union One Health 2021 Zoonoses Report. EFSA J. 2022 Dec 13;20(12):e07666. doi: 10.2903/j.efsa.2022.7666. PMID: 36524203; PMCID: PMC9745727.
109. Bullman S, Corcoran D, O'Leary J, O'Hare D, Lucey B, Sleator RD. Emerging dynamics of human campylobacteriosis in Southern Ireland. FEMS Immunol Med Microbiol. 2011 Nov;63(2):248-53. doi: 10.1111/j.1574-695X.2011.00847.x. PMID: 22077228.

3. A short paragraph concluding the noteworthy results is needed

A conclusion paragraph has been added to the manuscript.

Reviewer 2 Report

Comments and Suggestions for Authors

egarding the manuscript “Assessing the prevalence and dynamics of emerging Campylobacterales in human stool samples in Brussels by filtration culture”, the work is interesting and relevant in my opinion, the language used throughout the document is appropriate and clear, however I believe that section 2. Materials and methods could be further improved in terms of language accuracy and clarity. The author should ensure that the names of the microorganisms are italicised throughout the manuscript and that references to tables and figures are included in the text.

Line 18: “H2-enriched ” instead of “H2-enriched” 

Line 51: Since this is the first mention of the media, indicate the non-abbreviated designation. Thereafter, you can use the acronym.

Line 124: Identify the laboratory.

Line 156: identify the MALDI-TOF equipment used.

Line 208- Figure 1- Indicate the meaning of the coloured cells in the species and 

Mean % before COVID-19 and Mean % after COVID-20 columns.

Lines 273-362-The discussion of the optimal method for Campylobacter isolation by filtration is thorough; however, the hypotheses explaining why polycarbonate with a pore size of 0.6 μm is more suitable for Campylobacter isolation than cellulose with a pore size of 0.45 μm are not explored. It would be appropriate to present and discuss some hypotheses in this regard.

Line 479-480.  I don't understand the statement on lines 470-480. Why do you mention that with the use of PCR assays it is not possible to continue with the monitoring of antimicrobial resistance testing? Please justify this. 

Comments on the Quality of English Language

Language in Section 2 could be improved.

Author Response

Dear Editor,
We would like to thank you and the reviewers for your time and careful review of our work. We edited our manuscript following your recommendations. Responses to the reviewers are displayed below.
As requested, we provide enclosed the revised manuscript in two versions: one clean revised version and a version with track changes so that you can see where the changes have been made.

Regarding the manuscript “Assessing the prevalence and dynamics of emerging Campylobacterales in human stool samples in Brussels by filtration culture”, the work is interesting and relevant in my opinion, the language used throughout the document is appropriate and clear, however I believe that section 2. Materials and methods could be further improved in terms of language accuracy and clarity. The author should ensure that the names of the microorganisms are italicized throughout the manuscript and that references to tables and figures are included in the text.

Section 2. Materials and methods has been subject to a re-write. References to tables and figures

Line 18: “H2-enriched ” instead of “H2-enriched”

Correction has been made.

Line 51: Since this is the first mention of the media, indicate the non-abbreviated designation. Thereafter, you can use the acronym.

Non-abbreviated designations of media has been added to the manuscript: “Though, regarding C. jejuni and C. coli, comparison of isolation rates obtained by filtration culture and by culture on different selective media, i.e. Skirrow, Butzler, modified Charcoal-Cefoperazone-Deoxycholate agar (mCCDA) and Cefoperazone-Amphotericin B-Teicoplanin (CAT) media, showed non-significant [32,40,59,60,63,66] and contradictory [51,67] differences depending on the filters and selective media used.”

Line 124: Identify the laboratory.

The laboratory has been identified in the revised manuscript. This is the Laboratoire Hospitalier Universitaire de Bruxelles-Brussel Universitair Laboratorium (LHUB-ULB).

Line 156: identify the MALDI-TOF equipment used.

This has been clarified in the revised manuscript: “Matrix-assisted laser desorption ionisation-time of flight mass spectrometry (MALDI-TOF MS; Biotyper database, Bruker Daltonics, Bremen, Germany) was used to identify all growing colonies. The databases used are those used over time. Each new iteration was validated against a diverse panel of Campylobacter species in accordance with our quality assurance protocols.”

Line 208- Figure 1- Indicate the meaning of the colored cells in the species

Indication of the meaning of the colored cells has been added in the caption of Table 1: “Species prevalence of Campylobacterales in 51,065 stools at the Department of Microbiology, LHUB-ULB, Belgium, from 01/01/2017 to 31/12/2023 (pre- and post-COVID-19 periods on a green and blue background, respectively) using Butzler selective medium plus filtration culture with 0.6 μm polycarbonate filter and Columbia agar; percentage indicating the proportion of each species in the total number of strains of Campylobacterales in a year.”

Line 208- Figure 1- Mean % before COVID-19 and Mean % after COVID-20 columns.

Correction has been made.

Lines 273-362-The discussion of the optimal method for Campylobacter isolation by filtration is thorough; however, the hypotheses explaining why polycarbonate with a pore size of 0.6 μm is more suitable for Campylobacter isolation than cellulose with a pore size of 0.45 μm are not explored. It would be appropriate to present and discuss some hypotheses in this regard.

With regard to the hypothesis that explains why polycarbonate material is more suitable for the isolation of Campylobacter sp. than cellulose acetate, Nielsen et al. [59] suggested that the smooth, glassy surface of the polycarbonate filter would be more suitable for the penetration of motile Campylobacter sp. than
the rough surface of the cellulose acetate filter, which could catch random particles and block Campylobacter sp. from passing through the filter.

Regarding the hypothesis explaining why a 0.6 μm pore size filter is more suitable for isolating Campylobacter sp. than a 0.45 μm pore size filter, this could be explained by the fact that the 0.45 μm pore size is too small for motile Campylobacter sp., especially C. concisus, to efficiently penetrate the filter.

These explanations have been added to the discussion of the revised manuscript.

Line 479-480. I don't understand the statement on lines 470-480. Why do you mention that with the use of PCR assays it is not possible to continue with the monitoring of antimicrobial resistance testing? Please justify this.

Given that molecular multiplex panel currently does not offer information about antimicrobial resistance, we suggest that monitoring of antimicrobial resistance could only continue if additional testing, i.e. reflex culture with phenotypic antimicrobial susceptibility testing is performed. This paragraph has been re-worded in the conclusion section of the revised manuscript.

“Currently, a growing number of laboratories are diagnosing campylobacteriosis using commercial multiplex PCR assays, the efficacy of which remains to be evaluated in the context of such emerging species. Furthermore, these assays provide no information about antimicrobial resistance. It is therefore recommended that a routine reflex culture be carried out in the event of a positive multiplex PCR result. In the event of a negative result in a patient exhibiting persistent gastrointestinal symptoms with an unidentified etiology, a control stool sample may also be collected and submitted to a NRC that performs filtration culture”